# Electrophysiological and Neuropsychological Indices of Cognitive Dysfunction in Patients with Chronic Insomnia and Severe Benzodiazepine Use Disorder

**DOI:** 10.3390/brainsci13030375

**Published:** 2023-02-21

**Authors:** Anna Castelnovo, Silvia Miano, Raffaele Ferri, Alberto Raggi, Michelangelo Maestri, Valentina Bottasini, Matteo Anelli, Marco Zucconi, Vincenza Castronovo, Luigi Ferini-Strambi, Mauro Manconi

**Affiliations:** 1Sleep Medicine Unit, Neurocenter of Southern Switzerland, Ospedale Civico, Civic Hospital of Lugano, Via Tesserete 46, 6900 Lugano, Switzerland; 2Faculty of Biomedical Sciences, Università della Svizzera Italiana, 6900 Lugano, Switzerland; 3University Hospital of Psychiatry and Psychotherapy, University of Bern, 3000 Bern, Switzerland; 4Sleep Research Centre, Oasi Research Institute—IRCCS, 94018 Troina, Italy; 5Unit of Neurology, G.B. Morgagni–L. Pierantoni Hospital, 47121 Forlì, Italy; 6Department of Neurosciences, Neurological Clinic, University of Pisa, 56124 Pisa, Italy; 7Sleep Disorders Center, Vita-Salute San Raffaele University, 20132 Milan, Italy; 8Department of Neurology, Inselspital, Bern University Hospital, 3010 Bern, Switzerland

**Keywords:** chronic insomnia, executive functioning, long-term benzodiazepine use, mismatch negativity, Oddball P300, psychophysiology

## Abstract

Benzodiazepine (BDZ) misuse is a growing health problem, with 1–2% of patients under BDZ treatment meeting the criteria for use disorder or dependence. Although BDZ addiction potential has been known for decades, much remains unknown its effects on brain functions. The aim of this study was to assess the neuropsychological and neurophysiological profile of a group of chronic insomniacs taking long-term high doses of benzodiazepine. We recruited 17 consecutive patients admitted to our third-level Sleep Medicine Unit for drug discontinuation (7 males, mean age 49.2 ± 11.2 years, mean education 13.7 ± 3.9 years, mean daily diazepam-equivalent BDZ: 238.1 ± 84.5 mg) and 17 gender/age-matched healthy controls (7 males, mean age 46.8 ± 14.1 years, mean education 13.5 ± 4.5 years). We performed a full neuropsychological evaluation of all subjects and recorded their scalp event-related potentials (Mismatch-Passive Oddball-Paradigm and Active Oddball P300 Paradigm). Patients with chronic insomnia and BDZ use disorder showed a profound frontal lobe executive dysfunction with significant impairment in the cognitive flexibility domain, in face of a preserved working, short and long-term memory. In patients, P300 amplitude tended to be smaller, mainly over the frontal regions, compared to controls. BDZ use disorder has a severe cognitive impact on chronic insomnia patients. Long-term high-dose BDZ intake should be carefully evaluated and managed by clinicians in this specific patient population, especially in relation to risky activities.

## 1. Introduction

Chronic insomnia is one of the most frequent sleep disorders, affecting about 5–10% of the general adult population [1]. According to international diagnostic criteria [1,2], insomnia is defined as a persistent difficulty in initiating and/or maintaining sleep in association with negative cognitive and/or physical consequences. Insomnia persisting for longer than 3 months, occurring with a frequency higher than 3 times per week, is usually classified as chronic. Cognitive behavioral therapy for insomnia is recommended as the first-line treatment for chronic insomnia in adults of any age (strong recommendation, high-quality evidence) [3]. Benzodiazepines (BDZ) and BDZ receptor agonists are theoretically recommended only in the short-term treatment of insomnia (≤4 weeks; weak recommendation, moderate-quality evidence), according to the European guidelines for insomnia management [3]. While BDZ and non-BDZ hypnotics have overall low use disorder potential in the general population, the risk of tolerance, misuse and overuse among insomniacs should always be considered [4,5]. Patients often tend to use BDZ for a longer period than recommended, at higher dosages than prescribed, or even for symptoms other than insomnia, especially in the case of early-onset insomnia and higher state-anxiety [6]. It is estimated that approximately 2% of the patients with a BDZ prescription meet the criteria for use disorder or dependence, particularly those with a personal or family history of a substance use disorder [7].

Despite the clinical evidence of tolerance and use disorder, the impact of high-dose long-term use of BDZ for insomnia has been poorly investigated. While processing speed and other selective deficits appear to be impaired in all BDZ users [8], elderly patients taking high doses of BDZ may display a more global cognitive impairment [9]. From a neurophysiological standpoint, acute and sub-chronic administration of BDZ has been reported to influence our brain response to deviant stimuli, whether or not our attention is intentionally focused on those odd stimuli (mismatch negativity or MMN or active oddball P300 paradigm). BDZ reduces P300 amplitude to induce a more inconstant—and possibly BDZ-type dependent—prolongation in P300 latency [10,11,12,13]. A modulating influence of the GABA receptor activity on MMN generation has also been reported, with BDZ decreasing MMN amplitude [14,15]. In addition, insomnia per se may impact on subjective and objective measures of cognitive functions [16]. However, no study seems to have been conducted on event-related potentials (ERPs) in insomniacs taking high doses of BDZ.

Hence, patients with chronic insomnia and long-term high-dose BDZ treatment would theoretically carry the highest risk of cognitive deficits, especially in terms of prefrontal executive functions, as they combine the deleterious effect of long-term sleep deprivation with those of sedative medications. Although BDZs have a clear sleep-promoting effect [17], which might, theoretically, counteract some negative cognitive consequences of insomnia, in a previous study [18], we showed that the chronic intake of high doses of BDZ has a moderate effect on sleep architecture, and a profound impact on sleep microstructure with a marked depression of both electroencephalographic (EEG) slow-wave activity and of its physiological instability [19].

Here, we aimed to assess cognitive functioning measured by means of a complete neuropsychological battery and of ERP neurophysiological measures (P300 and MMN), in a unique population of middle-aged patients with chronic insomnia and a long-term high-dose use of BDZ.

## 2. Material and Methods

### 2.1. Patients

For this observational, cross-sectional investigation, we screened patients with chronic insomnia and on long-term high-dosage BDZ treatment, admitted for a program of drug discontinuation at the Sleep Disorder Centre of San Raffaele Hospital in Milan. Inclusion criteria were: (1) a diagnosis of substance use disorder according to the Statistical and Diagnostic Manual of Mental Disorders, 5th Edition (DSM-V), along with a diagnosis of chronic insomnia according to the International Classification of Sleep Disorders, 3rd edition (ICSD-3) [1,2]; (2) age between 18 and 65 years; (3) BDZ daily dose at least twice higher than the maximum recommended dosage; (4) BDZ use >3 months. Exclusion criteria were: (1) the diagnosis of a sleep disorder other than insomnia; (2) a respiratory disturbance index (RDI) <5 events/hour; (3) the diagnosis of any psychiatric disorder other than drug use disorder and insomnia; (4) a mini-mental state evaluation (MMSE) [20] >24; (5) any other neurological or medical condition that would affect the planned assessment. As the enrolled patients assumed a variety of different BDZ, in different associations and dosages, the equivalent dose of diazepam was calculated for each patient, according to the conversion scale by Bazire [21].

Healthy control subjects were drug-free, age and gender matched to the patient group, and screened to exclude major sleep, neurological and psychiatric disorders with a thorough semi-structured interview.

All study procedures were reviewed and approved by the local Vita-Salute San Raffaele University’s Ethical Committee. All participants provided written consent upon participation. All research activities were conducted in accordance with the Helsinki Declaration.

### 2.2. Procedures

All subjects were clinically assessed via a structured medical interview, a neurological examination, the Pittsburgh Sleep Quality Index (PSQI) [22], in order to systematically quantify subjective sleep quality and screen for sleep disturbances. Patients were evaluated at the beginning of their hospitalization, after one adaptation night and one VPSG recording. The morning after the second hospitalization night, we performed ERPs recordings and subsequently, an extensive neuropsychological evaluation. BDZs were gradually tapered off after completing these tests. Patients recruited for this study partially overlap with the patients presented in a previous paper on sleep macrostructure in patients with insomnia and high dosage long term BDZ use [18].

### 2.3. Neuropsychological Evaluation

After the MMSE screening, we performed a complete neuropsychological evaluation, designed to investigate a wide range of cognitive functions: the Cognitive Estimation Test, which measures general cognitive abilities [23], the Rey Auditory Verbal Learning Test to evaluate short and long term memory, as well as learning abilities [24]; the Raven’s Progressive Matrices (Coloured Progressive Matrices or CPM) to measure general human non-verbal intelligence and abstract reasoning [25], the Attentional Matrices test to evaluate visual selective attention [26]; the Forward and Backward Digit Span to assess verbal short-term and working memory [27]; the Corsi block tapping task to measure spatial short-term memory [28]; the Rey Osterrieth complex figure to assess visuo-constructional abilities [29]; the Verbal (phonemic and semantic) Fluency task to evaluate executive functions and verbal production abilities [28]; the Trial Making test, for executive function and cognitive set shifting (A and B) [30]. The administration of the entire battery took approximately 60–90 min.

### 2.4. MMN and P300 Paradigms

All recordings were performed at 10:00 a.m. During the experiment, subjects sat on a chair in an electrically and acoustically shielded room and were instructed to relax and to keep as still as possible during the test. The EEG montage included a cap with 32 scalp unipolar sensors placed according to the 10–20 system [31] and referred to linked mastoids, with the ground electrode at FPz. EEG was coupled with 4 electro-oculogram (EOG) sensors to detect vertical and horizontal eye-movements. Electrode impedance was kept lower than 5 KΩ. EEG signals were band-pass filtered at 0.5 Hz (6 dB/oct) to 30 Hz (24 dB/oct) and continuously digitized at a rate of 500 Hz.

ERPs were elicited using sequences of acoustic stimuli created and presented using the STIM2 software (Neuroscan Inc.) according the following paradigms.

*Mismatch (Passive Oddball) Paradigm*—Mismatch Negativity (MMN) represents an ERP component of auditory evoked potentials, reflecting the cortical response on rare and discernible deviances from acoustic repetitive stimuli, regardless of whether someone is paying attention to the sequence. MMN is thought to be the result of an automatic comparison process between the stored information on the standard ongoing stimulus and the information on the incoming deviant tone [32].

Subjects were informed that they would hear a sequence of tones and instructed to ignore them by concentrating on reading a self-selected book or magazine. Each subject was presented with 2 sequences of 300 stimuli. Each sequence was composed of standard tones (92%) with a 1000 Hz frequency, and of deviant tones (8%) with a 1100 Hz frequency. All stimuli had a duration of 50 ms, a rise/fall of 5 ms, an intensity of 75-dB sound pressure level, and were presented binaurally through intra-aura stimulators. Tones were presented in random order with a fixed interstimulus interval of 500 ms.

*Active Oddball P300 Paradigm*—This is the most studied component of ERPs. The oddball paradigm provides that the subject receives an infrequent acoustic target, on a background of frequently occurring standard stimuli [33]. P300 occurs when a mental or physical response to the target stimulus is required, ensuring that attention is being paid to the stimuli.

Subjects were informed that they would hear low-pitched tones interspersed occasionally with high-pitched tones and were required to silently count the high-pitched tones. They were told that at the end of the trial they would be asked how many high tones they had heard. Each subject was presented with two sequences of 300 stimuli. Each sequence was composed of standard tones (80%) with a 1000 Hz frequency, and of target tones (20%) with a 2000 Hz frequency. Recordings were retained if counting error was below 10%. All stimuli had a duration of 100 ms, a rise/fall of 10 ms, an intensity of 90-dB, and were presented binaurally through intra-aura stimulators in random order with a fixed interstimulus interval of 2 s.

### 2.5. ERPs Analysis

We extracted 700 ms epochs for the Mismatch (Passive Oddball) Paradigm and 1100 ms epochs for the Active Oddball P300 Paradigm—both starting 100 ms before each stimulus onset. The 100 ms window before the stimulus onset was used for automatic baseline correction. We then averaged separately EEG epochs for the standard and for deviant/target tones. Trials exceeding ±80 µV were automatically excluded from the averages, as well as trials containing excessive eye movements, blinks, and bursts of muscle activity, amplifier clipping, or other extra-cerebral artefacts.

In order to detect the MMN, we obtained “difference waveforms” by subtracting event-related potentials elicited by the deviant stimuli from event-related potentials elicited by standard stimuli. MMN was measured at Fz, Cz, Pz and defined as peak negativity at Fz (usually the lead with the largest peak for the MMN) during the 100–250 ms latency range in the passive conditions difference (deviant-standard) waveforms. P300 was measured at Fz, Cz, Pz and defined as the most positive peak within the response to rare stimuli in the latency window of 300–400 ms at Pz (usually the lead with the largest peak for the P300). The scalp location of the peaks on topographical maps was used to support their correct identification according to Edlund and Nichols [34].

### 2.6. Statistical Analysis

Statistical between-group comparisons were carried out by means of the nonparametric Mann–Whitney test for independent datasets. Alpha significance was set to *p* < 0.05. We corrected alpha for multiple comparisons using Bonferroni correction (alpha = 0.05/16 = 0.003 for the neuropsychological analysis and alpha = 0.05/10 = 0.005 for the neurophysiological analysis). Cohen’s d was used as a measure of effect size [35]. According to Cohen, 0.2 is indicative of a small effect, 0.5 of a medium and 0.8 of a large effect size. All statistical analyses were performed using the commercially available Statistica software package (StatSoft, Inc., Tulsa, OK, USA, 2001, STATISTICA data analysis software system, version 6).

## 3. Results

Seventeen patients (7 males, 10 females, mean age 49.2 ± 11.2 years, mean education 13.7 ± 3.9 years) and 17 matched controls (7 males, 10 females, mean age 46.8 ± 14.1 years, mean education 13.5 ± 4.5 years) completed the study. The two groups did not differ in terms of age (*p* > 0.05, Cohen’s d = −0.03) and education (*p* > 0.05, Cohen’s d = −0.12). The mean diazepam-equivalent BDZ daily amount taken by patients was 238.1 ± 84.5 mg. Patients reported taking BDZ for a mean duration of 9.4 yrs (±3.4 yrs), with possible changes of dosages and type of BDZ across the years. The included patients were treated with different types of BDZ, taken alone or in combination: lormetazepam (14 pts), delorazepam (13 pts), lorazepam (3 pts), ketazolam (2 pts), triazolam (2 pts), bromazepam (1 pts), brotizolam (1 pts), alprazolam (1 pts).

Table 1 reports the mean values of neuropsychological test scores in both groups. The patient and the control group did not differ significantly for age, gender or education (Table 1). Statistical differences between groups emerged for MMSE scores, the Cognitive Estimation Test, semantic and phonetic verbal fluency, trail making test, and Rey–Osterrieth complex figure Recall, as well as for the Pittsburgh Sleep Quality Index. In all these tests, long-term BDZ users performed significantly worse than controls (Table 1).

Figure 1 displays the grand averages of the MMN traces obtained in controls and long-term BDZ users from the midline EEG electrodes. Figure 2 shows the grand averages of the P300 traces obtained in controls and BDZ users from the midline EEG electrodes.

Table 2 shows the comparison between ERPs parameters obtained in the two groups and measured over the midline EEG electrodes. The amplitude of the MMN was lower in patients over the central and frontal midline EEG leads than in controls, but the difference between groups was not statistically significant. However, a large effect size was evident for the MMN amplitude over Fz. The amplitude of the P300 component of ERPs was significantly lower in patients compared to controls over all midline EEG leads. The difference between groups was large for Fz and Cz but did not survive Bonferroni’s correction for multiple comparisons. All latency measures of these two ERPs components were not significantly different between the two groups.

## 4. Discussion

We herein describe for the first time the neuropsychological and neurophysiological features of patients with chronic insomnia with no other major psychiatric comorbidity, who developed a long-term high-dose BDZ use, compared to healthy control subjects.

Of note, the population considered in our study is unique: our insomnia patients were taking doses of BDZ that were approximately 15 times higher than the maximum recommended daily dose, while the majority of previous studies considered subjects taking doses from 2 to 3 times higher than recommended.

### 4.1. Neuropsychological and Neuropsychological Tests Pointed to a Marked Frontal Executive Disfunction

Compared to healthy subjects, our sub-population of insomniacs displayed significantly lower MMSE scores, a widely used measure of general cognitive abilities. In addition, our group of patients showed a large and significant impairment in verbal ability (verbal semantic and phonemic fluency tests) [36], and in visual attention and task-switching (Trail Making Test-B).

Although the result did not survive multiple comparison adjustment, a large effect size was found for estimation abilities as measured by the Cognitive Estimation Test, which has been shown to be associated with frontal lobe injury and executive dysfunction [37]. Similarly, we found uncorrected-significance with a large effect size for Trail Making Test-A that assesses visual search and motor speed skills [38].

In line with the literature on BDZ effects in both animals and humans [39], we did not find a significant effect neither for verbal and visuo-spatial working memory (Digit Span and Corsi Block Test), nor for long-term verbal and visual memory (Rey Auditory Verbal Learning Test, Rey–Osterrieth Complex Figure Test). Finally, no difference was found between groups in terms of general intelligence (as measured by the Raven Matrices) and attention (Attentional Matrices).

Overall, these results point to a specific frontal lobe executive function impairment [40], especially in the cognitive flexibility and inhibition domain [38,41], as well as in processing speed. The implication of frontal lobes was also supported by our neurophysiological results. We found a trend towards a P300 amplitude reduction, more prominent over the frontal regions, as well as a high effect size in the (yet non-significant) difference in MMN amplitude over Fz. The scalp location of these effects mirrors our neuropsychological findings and reflects a subtle impairment of the automatic data processing in these areas, especially when active attention is required.

### 4.2. Comparison with the Previous Literature

Only one study explored the neuropsychological effect of high-dose BDZ [42] on a population with mean daily diazepam equivalent dose similar to ours (253.5 ± 221.6 mg). This study involved a group of patients admitted to an Addiction Unit with an established diagnosis of BDZ use disorder lasting for more than 6 months and daily BDZ intake at least five times the maximum daily recommended dose (i.e., >50 mg diazepam/day), without neurological comorbidities, concurrent alcohol or psychotropic drug dependence. However, insomnia or other sleep diagnoses were not mentioned, and anxiety and depression were not among the exclusion criteria. In line with our results, Trail Making Test B but not A, was significantly worse in patients than controls, as well as Stroop test, another cognitive test for the assessment of frontal lobe processing functions, the Symbol Digit Modalities Test, which assesses psychomotor (processing and motor) speed, and the Paced Auditory Serial Addition Test, that measures auditory information processing speed and flexibility.

In contrast with our findings, the Digit Span Forward and Backward tests were also different between groups. However, it has to be noted that these tests correlated with the severity of self-reported measures of anxiety and that anxiety is known to be reliably related to poorer performance on measures of working memory capacity [43]. Furthermore, explicit episodic memory as measured by the Verbal Paired Associates and visuospatial learning and memory as measured by the Visual-Spatial Memory and Recall Test were compromised in these patients, with the latter being influenced by the BDZ cumulative dose. Moreover, for these latter results, a possible bias of psychiatric comorbidities could not be ruled out, as working memory and long-term memory are usually compromised in depressed patients, persisting even after the remission of mayor depressive episodes and worsening with repeated episodes [44].

### 4.3. The Complex Interplay between Insomnia and BDZ

While some older studies on BDZ and Z-drug use reported an impact of these molecules on cognitive function, more recent reviews have highlighted the existence of mixed findings [45,46], regarding global cognitive functioning, psychomotor speed, and speed of processing [47]. A recent large longitudinal study showed no evidence that new or recurrent use of BDZ or Z-drug use could be associated with a significant decline in MMSE or in verbal fluency among the middle-aged population. However, these studies considered the impact of BDZ at standard therapeutical doses. Indeed, in line with our results, the BDZ effect on cognition seems to be rather duration- and/or dose-dependent, with a higher risk in the elderly [48].

In our group, the effect of insomnia per se in increasing the risk of cognitive impairment in treated patients could not be ruled out because individuals with insomnia often complain of subjective difficulties with concentration, attention and memory. While studies on community-based insomnia samples failed to detect greater objective cognitive impairment on formal testing, insomnia individuals who seek clinical treatment may be at a higher risk, especially elderly patients in the presence of other health problems and lower premorbid cognitive function [49,50,51]. Objective testing offered mixed results regarding cognitive impairments in attention, memory, and executive function [52]. There is limited evidence for an impairment in working memory, episodic memory and problem solving and some attentional processes, such as choice reaction time, information processing and selective attention, with preserved divided and sustained attention, perceptual and psychomotor processes, verbal functions, procedural memory and some aspects of executive functioning (verbal fluency, flexibility), as well as of general cognitive functioning [53]. However, these results suffer from possible biases related to limited statistical power (small sample size), inadequate matching for education or intellectual potential, confounding factors such as comorbid conditions (medical, psychological and other sleep disorders), the use of medications, and the heterogeneity intrinsic to chronic insomnia per se [53]. In this respect, patients with objective short sleep duration (<6 h) seem to display greater neuropsychological impairment (spatial span, fluency, brief visuospatial memory test, managing emotions, and continuous performance tests but not TMT-A, verbal learning and general cognition), compared to insomnia patients with normal sleep duration (brief visuospatial memory test and continuous performance tests) [54]. These data are consistent with the notion that the prefrontal cortex functioning is particularly vulnerable to sleep loss or disturbed sleep, both in insomnia as well as in related experimental conditions, such as sleep deprivation [55].

To conclude, the cognitive profile we described in our patients may be derived by the combined impact of insomnia and high-dose BDZ. However, the specific cognitive profile we described appears to have larger areas of overlap with the literature on the cognitive effects of BDZ. In support of this interpretation, a recent study pointed to an executive function impairment (TMT-A and categorical fluency) and a trend for lower MMSE and Digit Span Backward in patients affected by chronic insomnia under BDZ treatment, compared to drug-free patients and controls. Executive impairment was correlated with the duration of BDZ exposure [6]. Our neurophysiological data further support this interpretation. Indeed, P300 amplitude seems to increase in insomniacs after a bad night, as compared to controls [16,56], whereas in healthy volunteers, BDZs determine an attenuation of attention performance, with a decrease in P300 amplitude [57,58].

## 5. Limitations

Our sample reflects a unique condition, for the high dosage of BDZ consumption and the diagnosis of substance use disorder, and portraits a real-word scenario informative of clinical practice. However, some limitations must be taken into account. Given the specificity of our inclusion criteria, our sample size was relatively small, which limited our statistical power to detect significant differences between groups. Furthermore, considering that this was a cross-sectional and not a prospective study, and that we did not have a drug-free control insomnia group, or follow-up measures in our patients after our detox protocol, we could not offer a definitive answer to the question whether cognitive effects are mainly driven by high-dose BDZ use, chronic insomnia per se or a mixed effect of both. Last but not least, we could not perform a broad-spectrum objective evaluation of our patients. Thus, we could not exclude the possible impact of homeostatic and circadian factors on our cognitive and ERP variables.

## 6. Conclusions

Despite strong recommendations based on high-quality evidence warn about the potential adverse effects of BDZ use, particularly in the geriatric population, the long term-prescription of these medications and their inadequate use by patients is still widely diffuse [45].

Our current data highlight a specific frontal dysfunction in a unique population of middle-aged patients suffering from chronic insomnia who developed a long-lasting severe BDZ use disorder.

Although the majority of the studies are focused on the effect on the elderly, our study reinforces the idea that particular attention should be paid by clinicians to all age groups.

Future larger and prospective studies assessing neurocognitive and neurophysiological variables before and after BDZ discontinuation and including a control group of drug-free insomnia patients is warranted in order to disentangle the complex interplay between insomnia per se and medications.

## Figures and Tables

**Figure 1 brainsci-13-00375-f001:**
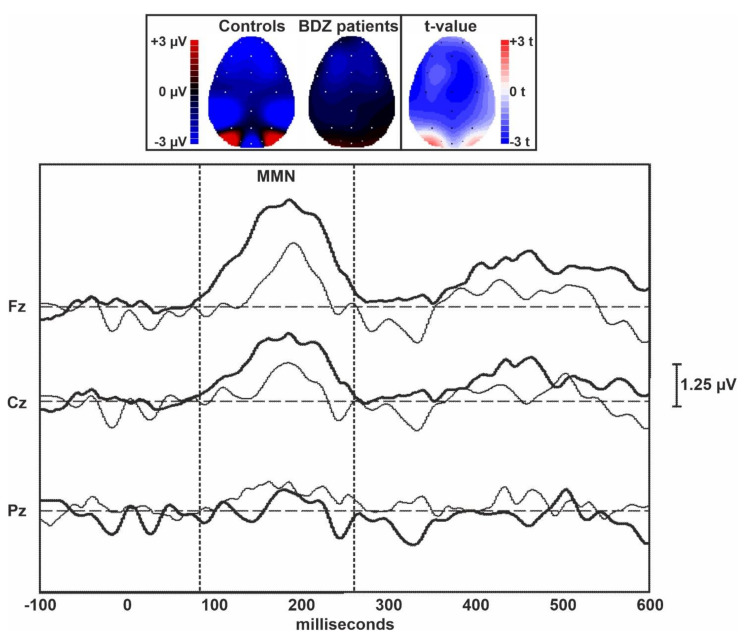
Topographic maps and grand averages of the MMN traces obtained in controls (thick lines) and BDZ users (thin lines) from the midline EEG electrodes. The two vertical dotted bars indicate the approximate extent of the MMN component of ERPs.

**Figure 2 brainsci-13-00375-f002:**
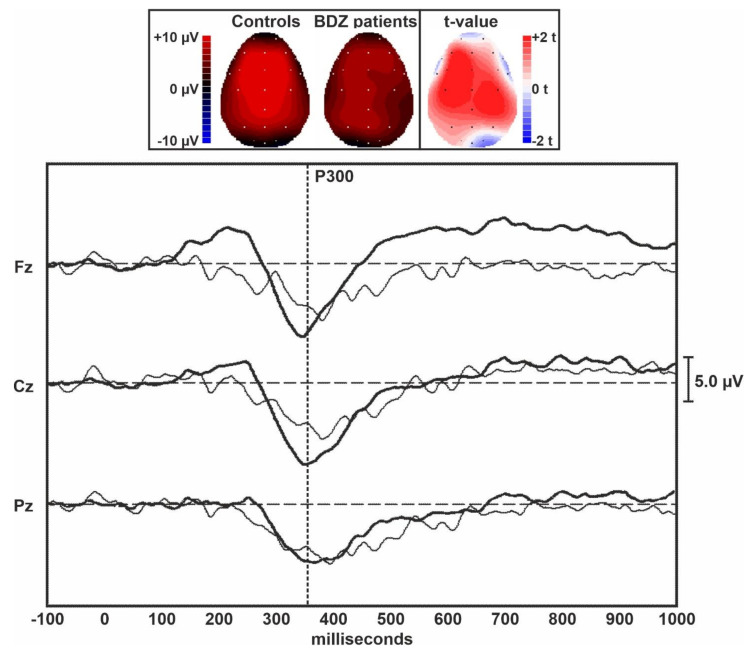
Topographic maps and grand averages of the P300 traces obtained in controls (thick lines) and BDZ users (thin lines) from the midline EEG electrodes. The vertical dotted bar indicates the approximate peak of the P300 component of ERPs.

**Table 1 brainsci-13-00375-t001:** Comparison between neuropsychological parameters.

	BDZ Users (*n* = 17)		Controls (*n* = 17)		Mann-Whitney	Effect Size
	Mean	S.D.	Mean	S.D.	*p*<	Cohen’s *d*
**Age, years**	46.1	11.65	46.8	14.09	NS	−0.034
**Education, years**	13.7	3.95	13.5	4.47	NS	−0.123
**Mini-mental state evaluation**	27.3	2.49	29.4	1.37	**0.00078 ***	**−1.052**
**Cognitive estimation test**	16.6	5.00	11.8	5.02	0.02	**0.970**
**Fluency, semantic**	38.8	7.54	50.0	7.13	**0.00024 ***	**−1.523**
**Fluency, phonological**	27.2	9.88	42.7	10.86	**0.00034 ***	**−1.496**
**Digit span, forward**	5.5	1.07	6.1	1.27	NS	−0.552
**Digit span, backward**	4.1	1.30	4.8	0.95	NS	−0.672
**Corsi block test**	4.6	0.93	5.2	0.81	NS	−0.607
**Rey auditory verbal learning test, learning**	41.4	8.49	48.1	14.77	0.02	−0.552
**Rey auditory verbal learning test, recall**	9.9	2.61	11.1	3.31	NS	−0.375
**Rey auditory verbal learning test, recognition**	12.9	1.52	14.6	3.14	NS	−0.686
**Attentional Matrices Test**	52.5	5.41	53.8	9.23	NS	−0.171
**Trail making test A (time)**	52.9	25.74	33.6	17.15	**0.0069**	**0.884**
**Trail making test B (time)**	145.8	73.96	75.5	44.92	**0.0014 ***	**1.149**
**Rey–Osterrieth complex figure test, copy**	31.5	4.49	33.7	2.40	NS	−0.613
**Rey–Osterrieth complex figure test, recall**	14.5	5.67	21.3	11.65	**0.023**	−0.736
**Raven matrices**	29.0	4.08	29.6	8.77	NS	−0.095
**Pittsburgh sleep quality index**	13.1	4.23	5.9	2.66	**0.00001 ***	**2.033**

Large effect sizes and significant (*p* < 0.05) differences are in bold characters. * Significant effects after Bonferroni correction.

**Table 2 brainsci-13-00375-t002:** Comparison between ERPs parameters.

	BDZ Users (*n* = 17)		Controls (*n* = 17)		Mann–Whitney	Effect Size
	mean	S.D.	mean	S.D.	*p*<	Cohen’s *d*
** *MMN* **						
**Fz latency, ms**	194.9	27.99	199.7	27.91	NS	−0.17
**Fz amplitude, µV**	−2.8	1.37	−3.8	1.10	NS	**0.826**
**Cz latency, ms**	194.3	28.43	200.3	28.35	NS	−0.214
**Cz amplitude, µV**	−2.2	0.73	−2.0	1.97	NS	−0.103
** *P300* **						
**Fz latency, ms**	357.4	32.65	347.6	29.65	NS	0.312
**Fz amplitude, µV**	5.9	2.49	11.3	6.05	**0.0095**	**−1.169**
**Cz latency, ms**	356.7	33.47	350.7	34.59	NS	0.176
**Cz amplitude, µV**	7.9	2.85	13.3	6.59	**0.0078**	**−1.067**
**Pz latency, ms**	357.7	35.50	351.5	34.69	NS	0.179
**Pz amplitude, µV**	8.2	2.72	11.0	5.42	**0.05**	−0.657

Large effect sizes and significant (*p* < 0.05) differences are in bold characters.

## Data Availability

The data presented in this study are available on request from the corresponding author.

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
