# Peer review of "Electrophysiological and Neuropsychological Indices of Cognitive Dysfunction in Patients with Chronic Insomnia and Severe Benzodiazepine Use Disorder"

_brainsci, 2023, doi:10.3390/brainsci13030375_

Round 1
Reviewer 1 Report
. For the section 2.2. Procedures, The authors should describe the study protocol and order in details. As the authors noted in page 3, lines 114 to 115, “the patients were still taking BDZs at the beginning of hospitalization”. Thus, the timings at taking BDZs and test session were critical for affecting the neuropsychological and neurophysiological profiles in patients.
- For the section 2.3. Neuropsychological evaluation, the authors noted in page 3, lines 128 to 129, “The administration of the entire battery took approximately 60 minutes”. This description means for me “all the tests were completed in less than 60 minutes”. If it is true, is the entire battery test (nine tests!!) a considerable load for subjects? And if this description means that “one of the tests was completed in less than 60 mins”, the study protocol and order should be written. Please mention this point.
- For the section 2, the detailed information in patients should be added. For instance, the ranges of age and dosage, the duration of administration (mean ± S.D. & range) and the drug name were needed, because these parameters could have a significant impact on the results.
- It is suspicious that the participants in the present study were similar to the author’s previous report (Manconi et al., 2017; doi: 10.1016/j.clinph.2017.03.009), because the average dose of benzodiazepines was exactly same. If the authors have used the same dataset, it would need to state this point explicitly. And the authors should re-check that the parameters in subjects were exactly correct.
- In the page 8, lines 240 to 243, the insomnia patients in the current study were taking doses of BDZ that were approximately 15 times higher than the maximum recommended daily dose. I think it is extraordinary. Please explain why these subjects were recruited (national character? regional medical care system?...).
Minor comments:
- in page 2, please add the brief descriptions about P300 amplitude and MMN for unfamiliar readers.
- in page 2, line 67, the abbreviation, “MMN” was first appeared. Please replace it to “Mismatch Negativity (MMN)” in page 4, line 142.
3) in page 8, Table 2, the data for MMN latency and amplitude over Pz were not represented. It is OK?
4) in the whole text, the abbreviations for benzodiazepines, “BDZ” and “BZD” were appeared. Please unify them.
I hope these comments will be helpful.
Author Response
Author's Notes
. For the section 2.2. Procedures, The authors should describe the study protocol and order in details. As the authors noted in page 3, lines 114 to 115, “the patients were still taking BDZs at the beginning of hospitalization”. Thus, the timings at taking BDZs and test session were critical for affecting the neuropsychological and neurophysiological profiles in patients.
We apologize for the lack of details provided to the reader. We added the following specifications: “Patients were evaluated at the beginning of their hospitalization, after one adaptation night and one VPSG recording. The morning after the second hospitalization night, we performed ERPs recordings and subsequently, an extensive neuropsychological evaluation. BZD were gradually tapered off after completing these tests.”
- For the section 2.3. Neuropsychological evaluation, the authors noted in page 3, lines 128 to 129, “The administration of the entire battery took approximately 60 minutes”. This description means for me “all the tests were completed in less than 60 minutes”. If it is true, is the entire battery test (nine tests!!) a considerable load for subjects? And if this description means that “one of the tests was completed in less than 60 mins”, the study protocol and order should be written. Please mention this point.
We meant that all the tests were completed in about 1 hour/maximum 1 hour and a half. As the estimation is approximate and some patients took longer than scheduled, we corrected 60 mins to 60/90 mins.
- For the section 2, the detailed information in patients should be added. For instance, the ranges of age and dosage, the duration of administration (mean ± S.D. & range) and the drug name were needed, because these parameters could have a significant impact on the results.
We agree with the reviewer. We opted to avoid to add an additional table in the text but we added the following statements in the text: “Patients reported to take BDZ for a mean duration of 9.4 yrs (±3.4 yrs), with possible changes of dosages and type of BDZ across the years. The included patients were treated with different type of BDZ, taken alone or in combination: lormetazepam (14 pts), delorazepam (13 pts), lorazepam (3 pts), ketazolam (2 pts), triazolam (2 pts), bromazepam (1 pts), brotizolam (1pts), alprazolam (1pts)”.
- It is suspicious that the participants in the present study were similar to the author’s previous report (Manconi et al., 2017; doi: 10.1016/j.clinph.2017.03.009), because the average dose of benzodiazepines was exactly same. If the authors have used the same dataset, it would need to state this point explicitly. And the authors should re-check that the parameters in subjects were exactly correct.
We are deeply grateful to the reviewer for the nice catch. The reported dosages were a misprint from the previous paper. The mean diazepam-equivalent BZD daily dosage for the patients that participated to the current study (that partially overlap with the previous) was 238.1 ±84.5 mg.
We also added a statement to specify that part of our population partially overlap with the patients that were included in the study on sleep macrostructure by Manconi et al. “Patients recruited for this study partially overlap with the patients presented in a previous paper on sleep macrostructure in patients with insomnia and high dosage long term BDZ use [18].” Sadly, not all patients agreed to perform ERPs and neuropsychological tests.
- In the page 8, lines 240 to 243, the insomnia patients in the current study were taking doses of BDZ that were approximately 15 times higher than the maximum recommended daily dose. I think it is extraordinary. Please explain why these subjects were recruited (national character? regional medical care system?...).
These patients were recruited via the regional medical care system. As San Raffaele Hospital is one of the few national centers to offer this service, severe cases were recruited from all across Italy. Only patients with severe abuse were offered the possibility of a hospitalization period for drug discontinuation in our Sleep Unit.
Minor comments:
- in page 2, please add the brief descriptions about P300 amplitude and MMN for unfamiliar readers.
We agree with the reviewer that some readers might not be familiar with ERPs. Therefore, we embedded a short explanation about P300 and MMN within the introduction, as suggested by the reviewer: “From a neurophysiological standpoint, acute and sub-chronic administration of BDZ has been reported to influence brain response to deviant stimuli, whether or not attention is intentionally focused on infrequent stimuli (mismatch negativity or MMN or active oddball P300 paradigm).”
We prefer to avoid more details in the introduction, as this topic is more extensively covered in the subsequent Method Section.
- in page 2, line 67, the abbreviation, “MMN” was first appeared. Please replace it to “Mismatch Negativity (MMN)” in page 4, line 142.
We thank the reviewer for the catch. We have now added “Mismatch Negativity” at the first appearance of the abbreviation MMN (see minor comments, point 1).
3) in page 8, Table 2, the data for MMN latency and amplitude over Pz were not represented. It is OK?
We did not include Pz because the auditory MMN is a fronto-central negative potential with source in the primary and non-primary auditory cortex and a typical latency of 150-250 ms after the onset of the deviant stimulus.
4) in the whole text, the abbreviations for benzodiazepines, “BDZ” and “BZD” were appeared. Please unify them.
We replaced BZD with BDZ whenever it occurred in the text.
I hope these comments will be helpful.
Thank you very much for your useful comments.
Reviewer 2 Report
In this paper the authors investigated cognitive dysfunction in patients with chronic insomnia, associated with severe benzodiazepine use disorder and found profound frontal lobe executive dysfunction with significant impairment in the cognitive flexibility domain while preserved working, short- and long-term memory.
There are several crucial issues that are suggested to be clarified:
-
In the Materials and Methods, L.91, the authors indicated an inclusion criterion: “...daily dose at least twice higher than the maximum recommended dosage”; in Discussion, L.239, however, they refer to the fact that “... our insomnia patients were taking doses of BDZ that were approximately 15 times higher than the maximum recommended daily dose”. What could have caused inclusion of patients with doses much higher than inclusion threshold? What was the mean and dispersion of this factor?
-
Have the authors performed regression analyses on these 17 patients for testing correlations between investigated cognitive functions and BZN dosing, and duration of BZN use?
-
It is indicated that control and BZN groups were similar in age and gender, however, did these cofactors had effect within these groups?
-
Have the authors checked whether investigated cognitive functions could be age or gender-related?
-
I wonder, whether the studied effects of benzodiazepine could have depend on individual circadian clock properties, or habitual sleep preferences (chronotype)? Also, it is important to know how much different was sleep duration and sleep phase in BZN groups.
-
It is interesting to know, whether impairments of cognitive functions in this study were related to differences in sleep duration and sleep phase.
Author Response
Author's Notes
- In the Materials and Methods, L.91, the authors indicated an inclusion criterion: “...daily dose at least twice higher than the maximum recommended dosage”; in Discussion, L.239, however, they refer to the fact that “... our insomnia patients were taking doses of BDZ that were approximately 15 times higher than the maximum recommended daily dose”. What could have caused inclusion of patients with doses much higher than inclusion threshold? What was the mean and dispersion of this factor?
Our patients were recruited through the regional medical care system. As San Raffaele Hospital was one of the few hospitals to offer hospitalization for BDZ tapering, patients came from all around Italy. Only patients with severe abuse were offered the possibility of a hospitalization period for drug discontinuation in our Sleep Unit. Corrected data concerning the mean dose of BDZ taken at the moment of the evaluation, as well as its SD, have been provided in the results section: The mean diazepam-equivalent BDZ daily amount taken by patients was 238.1 ±84.5 mg mg.”
- Have the authors performed regression analyses on these 17 patients for testing correlations between investigated cognitive functions and BZN dosing, and duration of BZN use?
Sadly, we did not perform a regression analysis. The duration of BDZ use is difficult to quantify. This is an historical data collected by the patient, but across the years of treatment, the patients changed the dose and the type of BDZ several times, making difficult and less reliable the analysis.
- It is indicated that control and BZN groups were similar in age and gender, however, did these cofactors had effect within these groups? Have the authors checked whether investigated cognitive functions could be age or gender-related?
Yes, of course, age, gender and education may affect the majority of the cognitive tests that we analyzed. These variables were taken into account when correcting the final scores of the tests, when correction was available. Of note, there was no difference between age, gender and education between groups.
- I wonder, whether the studied effects of benzodiazepine could have depend on individual circadian clock properties, or habitual sleep preferences (chronotype)? Also, it is important to know how much different was sleep duration and sleep phase in BZN groups.
It is interesting to know, whether impairments of cognitive functions in this study were related to differences in sleep duration and sleep phase.
We fully agree with the reviewer that this is an interesting point, but while none of our patients had a clinical diagnosis of circadian sleep disorder, or of insufficient sleep syndrome, neither was recognized as short or long sleeper (< 5 or > 10 hours of sleep per day), we do not have objective markers regarding sleep duration and sleep phase in our patient group. Thus, we listed this important aspect among the limitations of the current study, as follows: “Last but not least, we could not exclude the possible impact of homeostatic and circadian factors on our cognitive and ERP variables.”
Round 2
Reviewer 1 Report
The manuscript has been revised well. There is only one minor correction that the words for mean of BDZ doses, “223.9±312.2 mg” should be corrected to “238.1±84.5 mg” in the Abstract section. If this part of the correction is completed, I think this manuscript will be acceptable.